# Deep Ensemble Bayesian Active Learning : Adressing the Mode Collapse issue in Monte Carlo dropout via Ensembles

## Abstract

In image classification tasks, the ability of deep convolutional neural networks (CNNs) to deal with complex image data has proved to be unrivalled. Deep CNNs, however, require large amounts of labeled training data to reach their full potential. In specialized domains such as healthcare, labeled data can be difficult and expensive to obtain. One way to alleviate this problem is to rely on active learning, a learning technique that aims to reduce the amount of labelled data needed for a specific task while still delivering satisfactory performance. We propose a new active learning strategy designed for deep neural networks. This method improves upon the current state-of-the-art deep Bayesian active learning method, which suffers from the mode collapse problem. We correct for this deficiency by making use of the expressive power and statistical properties of model ensembles. Our proposed method manages to capture superior data uncertainty, which translates into improved classification performance. We demonstrate empirically that our ensemble method yields faster convergence of CNNs trained on the MNIST and CIFAR-10 datasets.

## 1 Introduction

The success of deep learning in the last decade has been attributed to more computational power, better algorithms and larger datasets. In object classification tasks, CNNs widely outperform alternative methods in benchmark datasets (LeCun et al., 2015) and have been used in medical imaging for critical situations such as skin cancer detection (Haenssle et al., 2018), retinal disease detection (De Fauw et al., 2018) or even brain tumour survival prediction (Lao et al., 2017).

Although their performance is unrivalled, their success strongly depends on huge amounts of annotated data (Bengio et al., 2007; Krizhevsky et al., 2012). In specialized domains such as medicine or chemistry, expert labelled data is costly and time consuming to acquire (Hoi et al., 2006; Smith et al., 2018). Active Learning (AL) provides a theoretically sound framework (Cohn et al., 1996) that reduces the amount of labelled data needed for a specific task. Developed as an iterative process, AL progressively adds unlabelled data points to the training set using an acquisition function, ranking them in order of importance to maximize performance.

Using Active Learning within a Deep Learning framework (DAL) has recently seen successful applications in text classification (Zhang et al., 2017; Shen et al., 2017), visual question answering (Lin & Parikh, 2017) and image classification with CNNs (Gal et al., 2017; Sener & Savarese, 2017; Beluch et al., 2018). One key difference between DAL and classical AL is the sampling in batches, which is needed to keep computational costs low. As such, developing scalable DAL methods for CNNs presents challenging problems. Firstly, acquisitions functions do not scale well for high dimensional data or parameter spaces, due to the cost of estimating uncertainty measures, which is the main approach. Secondly, even with scalability not being an issue, one needs to obtain good uncertainty estimates in order to avoid having overconfident predictions. One of the most promising techniques is Deep Bayesian Active Learning (DBAL) (Gal, 2016; Gal & Ghahramani, 2016), which uses Monte-Carlo dropout (MC-dropout) as a Bayesian framework to obtain uncertainty estimates. However, as mentioned in Ducoffe & Precioso (2018), uncertainty-based methods can be fooled by adversarial examples, where small perturbations in inputs can result in overconfident and surprising

outputs. Another approach presented by Beluch et al. (2018) uses ensemble models to obtain better uncertainty estimates than DBAL methods, although there are no result on how it deals with adversarial perturbations. Whereas uncertainty-based methods aim to pick data points the model is most uncertain about, density-based approaches try to identify the samples that are most representative of the entire unlabelled set, albeit at a computational cost (Sener & Savarese, 2017). Hybrid methods aim to trade uncertainty for representativeness. Our belief is that overconfident predictions for DBAL methods are an outcome of the mode collapse phenomenon in variational inference methods (Srivastava et al., 2017), and that by combining the expressive power of ensemble methods with MC-dropout we can obtain "better" uncertainties without trading representativeness.

In this paper we provide evidence for the mode collapse phenomenon in the form of a highly imbalanced training set acquired during AL with MC-dropout, and show that 'preferential' behaviour is not beneficial for the AL process. Furthermore, we link the mode collapse phenomenon to overconfident classifications. We compare the use of ensemble models to MC-Dropout for uncertainty estimation and give intuitive reasons why combining the two might perform better. We present Deep Ensemble Bayesian Active Learning (DEBAL) which confirms our intuition for experiments on MNIST and CIFAR-10.

In Section 2 we give an overview of current popular methods for DAL. In Section 3, various acquisition functions are introduced and the mode collapse issue is empirically identified. Further on, the use of model ensembles is motivated before presenting our method DEBAL. The last part of section 3 is devoted to understanding the cause of the observed improvements in performance.

## 2 BACKGROUND

The area of active learning has been studied extensively before (see Settles (2012) for a comprehensive review), but with the emergence of deep learning, it has seen widespread interest. As proved by Dasgupta (2005) there is no good universal AL strategy, researchers instead relying on heuristics tailored for their particular tasks.

**Uncertainty-based Methods.** We identify *uncertainty-based methods* as being the main ones used by the image classification community. Deep Bayesian Active Learning (Gal et al., 2017) models a Gaussian prior over the CNNs weights and uses variational inference techniques to obtain a posterior distribution over the network's predictions, using these samples as a measure of uncertainty and as input to the acquisition function of the AL process. In practice, posterior samples are obtained using Monte-Carlo dropout (MC-dropout)(Srivastava et al., 2014), a computationally inexpensive and powerful stochastic regularization technique that performs well on real-world datasets (Leibig et al., 2017; Kendall et al., 2015) and has been shown to be equivalent to performing variational inference (Gal & Ghahramani, 2016). However, these approximating methods suffer from *mode collapse*, as evidenced in Blei et al. (2017). Another method, Cost-Effective Active Learning (CEAL) (Wang et al., 2016), uses the entropy of the network's outputs to quantify uncertainty, with additional pseudo-labelling. This can be seen as the deterministic counterpart of DBAL, that adds highly confident samples directly from predictions, without the query process. Käding et al. (2016) propose a method on the expected model output change principle. This method approximates the expected reduction in the model's error to avoid selecting redundant queries, albeit at a computational cost. Lastly, as this work was being developed, we found the work of Beluch et al. (2018), who propose to use deterministic ensemble models to obtain uncertainty approximations. Their method scores high both in terms of performance and robustness.

**Density-based Methods & Hybrid Methods.** Sener & Savarese (2017) looked at the data selection process from a set theory approach (core set) and showed their heuristic-free method outperforms existing uncertainty-based ones. Their acquisition function uses the geometry in the data-space to select the most informative samples. The main idea is to try to find a diverse subset of the entire input data space that best represents it. Although achieving promising results, the core set approach is computationally expensive as it requires solving a mixed integer programming optimisation problem. Ducoffe & Precioso (2018), on the other hand, rely on adversarial perturbation to select unlabeled samples. Their approach can be seen as margin based active learning, whereby distances to decision boundaries are approximated by distances to adversarial examples. To the best of our knowledge, the only hybrid method (combining measures of both uncertainty and representativeness) tested within a CNN-based DAL framework is the one proposed in Wang & Ye (2015).

Although originally not tested on CNNs, this method was shown to perform worse than the core set approach in Sener & Savarese (2017).

**Deep Bayesian Active Learning.**     Given the set of inputs $\mathbb{X} = \{\boldsymbol{x}_1, .., \boldsymbol{x}_n\}$ and outputs $\mathbb{Y} = \{y_1, .., y_n\}$ belonging to classes $c$, one can define a probabilistic neural network by defining a model $f(\boldsymbol{x}; \boldsymbol{\theta})$ with a prior $p(\boldsymbol{\theta})$ over the parameter space $\boldsymbol{\theta}$, usually Gaussian, and a likelihood $p(y = c|\boldsymbol{x}, \boldsymbol{\theta})$ which is usually given by $\mathrm{softmax}(f(\boldsymbol{x}; \boldsymbol{\theta}))$. The goal is to obtain the posterior distribution over $\boldsymbol{\theta}$:

$$p(\boldsymbol{\theta}|\mathbb{X}, \mathbb{Y}) = \frac{p(\mathbb{Y}|\mathbb{X}, \boldsymbol{\theta})p(\boldsymbol{\theta})}{p(\mathbb{Y}|\mathbb{X})} \tag{1}$$

One can make predictions $y^*$ about new data points $\boldsymbol{x}^*$ by taking a weighted average of the forecasts obtained using all possible values of the parameters $\boldsymbol{\theta}$, weighted by the posterior probability of each parameter:

$$p(y^*|\mathbf{x}^*, \mathbb{X}, \mathbb{Y}) = \int p(y^*|\boldsymbol{x}, \boldsymbol{\theta})p(\boldsymbol{\theta}|\mathbb{X}, \mathbb{Y})d\boldsymbol{\theta} = \mathbb{E}_{\boldsymbol{\theta} \sim p(\boldsymbol{\theta}|\mathbb{X}, \mathbb{Y})}[f(\boldsymbol{x}; \boldsymbol{\theta})] \tag{2}$$

The real difficulty arises when trying to compute these expectations, as has been previously covered in the literature (Neal, 2012; Hinton & Van Camp, 1993; Barber & Bishop, 1998; Lawrence, 2001). One way to circumvent this issue is to use Monte Carlo (MC) techniques (Hoffman et al., 2013; Paisley et al., 2012; Kingma & Welling, 2013), which approximate the exact expectations using averages over finite independent samples from the posterior predictive distribution (Robert & Casella, 2013). The MC-Dropout technique (Srivastava et al., 2014) will replace $p(\boldsymbol{\theta}|\mathbb{X}, \mathbb{Y})$ with the dropout distribution $\hat{q}(\boldsymbol{\theta})$. This method scales well to high dimensional data, it is highly flexible to accommodate complex models and it is extremely applicable to existing neural network architectures, as well as easy to use.

In DBAL (Gal, 2016), the authors incorporate Bayesian uncertainty via MC-dropout and use acquisition functions that originate from information theory to try and capture two types of uncertainty: *epistemic* and *aleatoric* (Smith & Gal, 2018; Depeweg et al., 2017). Epistemic uncertainty is a consequence of insufficient learning of model parameters due to lack of data, leading to broad posteriors. On the other hand, aleatoric uncertainty arises due to the genuine stochasticity in the data (noise) and always leads to predictions with high uncertainty. We briefly describe the three main types of acquisition functions:

- *MaxEntropy* (Shannon, 2001). The higher the entropy of the predictive distribution, the more uncertain the model is:

$$H[y|\boldsymbol{x}, \boldsymbol{\theta}] = -\sum_c p(y = c|\boldsymbol{x}, \boldsymbol{\theta})\mathrm{log}p(y = c|\boldsymbol{x}, \boldsymbol{\theta}) \tag{3}$$

- *Bayesian Active Learning by Disagreement (BALD)* (Houlsby et al., 2011). Based on the mutual information between the input data and posterior, and quantifies the information gain about the model parameters if the correct label would be provided.

$$I(y, \boldsymbol{\theta}|\boldsymbol{x}, \boldsymbol{\theta}) = H[y|\boldsymbol{x}; \mathbb{X}, \mathbb{Y}] - \mathbb{E}_{\boldsymbol{\theta} \sim p(\boldsymbol{\theta}|\mathbb{X}, \mathbb{Y})}\Big[H[\boldsymbol{y}|\boldsymbol{x}, \boldsymbol{\theta}]\Big] \tag{4}$$

- *Variation Ratio* (Freeman, 1965). Measures the statistical dispersion of a categorical variable, with larger values indicating higher uncertainty:

$$\mathrm{VarRatio}(\boldsymbol{x}) = 1 - \max_y p(y|\boldsymbol{x}, \boldsymbol{\theta}) \tag{5}$$

As seen in Gal (2016), for the above deterministic acquisition functions we can write the stochastic versions using the Bayesian MC-Dropout framework, where the class conditional probability $p(y|\boldsymbol{x}, \boldsymbol{\theta})$ can be approximated by the average over the MC-Dropout forward passes. The stochastic predictive entropy becomes:

$$H[y|\boldsymbol{x}, \boldsymbol{\theta}] = -\sum_c \Big(\frac{1}{K}\sum_k p(y = c|\boldsymbol{x}, \boldsymbol{\theta}_k)\Big)\mathrm{log}\Big(\frac{1}{K}\sum_k p(y = c|\boldsymbol{x}, \boldsymbol{\theta}_k)\Big) \tag{6}$$

K corresponds to the total number of MC-Dropout forward passes at test time. Equivalent stochastic versions can be obtained for all other acquisition functions.

Table 1: Experiment settings for **MNIST** and **CIFAR-10**

| Dataset | Model | Training epochs | Data size pool/val/test | Acquisition size |
|---------|-------|-----------------|-------------------------|------------------|
| MNIST | 2-Conv | 500 | 59,780 / 200 / 10,000 | 20 + 10 —>1,000 |
| CIFAR-10 | 4-Conv | 500 | 47,800 / 2000 / 10,000 | 200 + 100 —>10,000 |

## 3 DEBAL: DEEP ENSEMBLE BAYESIAN ACTIVE LEARNING

### 3.1 EXPERIMENTAL DETAILS

We consider the multiclass image classification task on two well-studied datasets: **MNIST** (LeCun, 1998) and **CIFAR-10** (Krizhevsky & Hinton, 2009). Table 1 contains a summary of the results, with the acquisition size containing the initial training set with the batch size for one iteration up to the maximum number of points acquired. At each acquisition step, a fixed sample set from the unlabelled pool is added to the initial balanced labelled data set and models are re-trained from the entire training set. We evaluate the model on the dataset's standard test set. The CNN model architecture is the same as in the Keras CNN implementation for MNIST and CIFAR-10 (Chollet et al., 2015). We use Glorot initialization for weights, Adam optimizer and early stopping with patience of 15 epochs, for a maximum of 500 epochs. We select the best performing model during the patience duration. We use MC-Dropout with $K = 100$ forward passes for the stochastic acquisition functions. In all experiments results are averaged over three repetitions. For the ensemble models discussed later, each ensemble consists of $M = 3$ networks of identical architecture but different random initializations.

### 3.2 EVIDENCE OF MODE COLLAPSE IN DBAL

Our experimental results confirmed the performance of Gal (2016) using MC-dropout. However, we observe a lack of diversity in the data acquired during the AL process. This effect is more extreme in the initial phase, which is an important factor when dealing with a small dataset classification problem (see Figure 1).

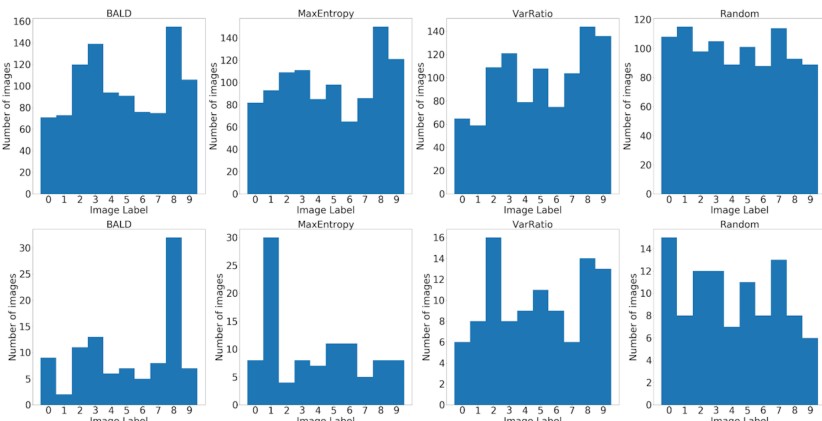

Figure 1: **MNIST** histograms of true labels in the training set. **Top:** End of AL process. Total number of images in training set: 1,000. **Bottom:** After first 8 acquisition iterations. Total number of images in training set: 100.

One can argue that the preferential behaviour observed is a desirable one and is arising from the fact that images belonging to some specific classes are more uncertain and difficult to classify, due to resemblance of data from other classes. To debunk this hypothesis, we trained a model with the same architecture on the entire 60,000 sample training set available and used this model to rank the

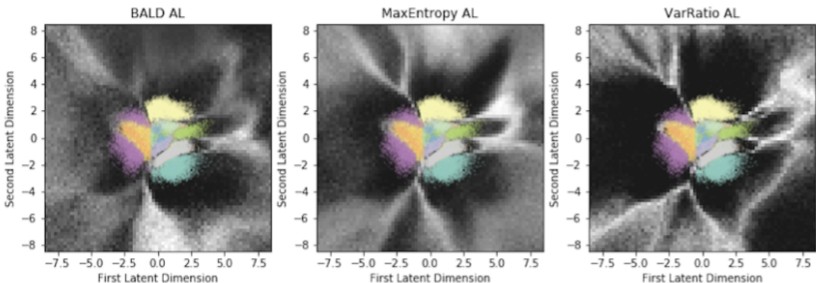

Figure 2: **MNIST** uncertainty visualization in VAE space at the end of the AL process for all measures. Colours represent different classes. Low uncertainty: black, High uncertainty: white.

uncertainty for each sample from the 10,000 samples test set. As can be seen in Appendix Figure 7, over-represented class labels during the AL experiment do not have high uncertainty. To assess positive effects of over-sampling, we evaluated how easy models at the end of AL process classified them and observed no such effect (Appendix Figure 8).

Smith & Gal (2018) argue that the MC-Dropout technique suffers from over-confident predictions. They are particularly concerned with the interpolation behaviour of the uncertainty methods across unknown regions (regions of the input space not seen during the model training phase). We performed a similar analysis in order to gain understanding into how these methods behave. Following their experimental setting, we use a VAE (Kingma & Welling, 2013) to project the MNIST dataset into a 2-dimensional latent space. Figure 2 allows us to visualize the encodings and decode points from latent space to image space, together with their associated measures of uncertainty. The large black regions behind the data suggest that the model is unrealistically over-confident about data that does not resemble anything seen during training, thus providing further evidence supporting MC-Dropouts main deficiency: mode-collapse.

### 3.3 DEEP ENSEMBLES: A RECIPE FOR USEFUL UNCERTAINTY ESTIMATION

We hypothesize that one of DBALs main deficiencies is its inability to capture the full posterior distribution of the data (mode collapse). This can prevent the model from learning in an optimal way, leading to unsatisfactory performance when classifying previously unseen images. As suggested in Smith & Gal (2018) and Lakshminarayanan et al. (2017), one intuitive fix would be to replace the single MC-Dropout model in the AL process with an *ensemble of MC-Dropout models*, with each member of the ensemble using a different initialization. Since one MC-Dropout model collapses around a subspace (one, or a few local modes) of the posterior distribution, a collection of such models, starting from different initial configurations, will end up covering different (and somehow overlapping) sub-regions of the probability density space. However, one key assumption here, is that each model member of the ensemble will end up capturing the behaviour around a different local mode. Beluch et al. (2018) test this idea in a deterministic setting, where the uncertainty resulting from the use of a deterministic ensemble proved to be more useful for the active learner than the uncertainty provided from a *single* MC-Dropout network.

We propose *DEBAL*, a stochastic ensemble of $M$ MC-Dropout models, with $M << K$. Each member of the ensemble is characterized by a different set of weights $\boldsymbol{\theta}_m$. We use the randomization-based approach to ensembles (Breiman, 1996), where each member of the ensemble is trained in parallel without any interaction with the other members of the ensemble. We consider the ensemble as a mixture model where each member of the ensemble is uniformly weighted at prediction time. For our task, this corresponds to averaging the predictions as follows:

$$p(y|\mathbf{x}; \mathbb{X}, \mathbb{Y}) = \frac{1}{M} \sum_m p(y|\boldsymbol{x}, \boldsymbol{\theta}_m) \tag{7}$$

---

**Algorithm 1** DEBAL: Deep Ensemble Bayesian Active Learning

---

**Input:** $\mathcal{L}$ - initial labeled training set, $\mathcal{U}$ - initial unlabeled training set, $\mathcal{H}$ - initial set of hyper-parameters to train the network, $acq.fn.$ - acquisition function, $n_{query}$ - query batch size, $N$ - final training set size, $K$ - number of forward passes in MC-Dropout, $M$ - number of models in the ensemble

**Initialize:**
    i=0, $\mathcal{L} \leftarrow \mathcal{L}_0, \mathcal{U} \leftarrow \mathcal{U}_0$

**while** $i < N$ **do**
    Train the ensemble members $A_{m,i}(m \in M)$ given the current labeled training set
    $A_{m,i} = training(\mathcal{H}, \mathcal{L}_i)$
    Form ensemble model $E_i$ = ensemble($A_1$,$A_2$,...,$A_M$)
    **for** $x_j \in \mathcal{U}$ **do**
        Compute uncertainty using the ensemble and MC-Dropout
        $r_j \leftarrow acq.fn.(x_j, E_i; K)$
    **end for**
    Query the labels of the $n_{query}^{\text{th}}$ samples $\mathcal{Q}_j$ with the largest uncertainty values
    $index_j \leftarrow argsort(r_j; n_{query})$
    $\mathcal{Q}_j \leftarrow \{x_z | z \in index_j[0 : n_{query}]\}$
    $\mathcal{L}_{i+1} \leftarrow \mathcal{L}_i \cup \mathcal{Q}_j$
    $\mathcal{U}_{i+1} \leftarrow \mathcal{U}_i \setminus \mathcal{Q}_j$
**end while**

---

Equation 7 corresponds to the *deterministic ensemble* case. Our predictions are further averaged by a number of MC-Dropout forward passes, giving rise to what we call a *stochastic ensemble*:

$$p(y|\boldsymbol{x}; \mathbb{X}, \mathbb{Y}) = \frac{1}{M}\frac{1}{K}\sum_m \sum_k p(y|\boldsymbol{x}, \boldsymbol{\theta}_{m,k}) \tag{8}$$

$\boldsymbol{\theta}_{m,k}$ denotes the model parameters for ensemble model member $m$ in the $k$ MC-Dropout forward pass. Each of the two equations can then be used with acquisitions functions previously described. In the *deterministic ensemble* case, we just replace the number of forward passes $k$ with the number of ensemble classifiers $m$ to obtain expressions for uncertainty. The predictive entropy for our *stochastic ensemble* becomes:

$$\mathrm{H}[y|\boldsymbol{x}; \mathbb{X}, \mathbb{Y}] = -\sum_c \Big(\frac{1}{M}\frac{1}{K}\sum_m \sum_k p(y = c|\boldsymbol{x}, \boldsymbol{\theta}_{m,k})\Big) \log\Big(\frac{1}{M}\frac{1}{K}\sum_m \sum_k p(y|\boldsymbol{x}, \boldsymbol{\theta}_{m,k})\Big) \tag{9}$$

For both datasets, DEBAL shows significant improvements in classification accuracy (Figure 3 - similar results obtained for all other acquisition functions but for sake of clarity we illustrate results for BALD only). The better performance of the deterministic ensemble method over the single MC-Dropout one is in agreement with similar results presented in Beluch et al. (2018), and is attributed to better uncertainty estimates obtained from the ensemble. We hypothesize that the additional improvement is a result of better uncertainty estimates from the stochastic ensemble.

To validate our claims, we compare the uncertainty behaviour between single network MC-dropout and DEBAL, as can be seen qualitatively in Appendix Figure 10 by the elimination of "black holes" in the latent space of DEBAL. Secondly, we observe how the methods behave on both seen and unseen distributions, using the **NotMNIST** dataset of letters A-J from different fonts (Bulatov, 2011). BALD uncertainty results for this approach are evidenced in Figure 4. We sample 2,000 balanced and random images from the MNIST test set and, similarly, 2,000 images from the NotMNIST test set. For MNIST, we make sure that the randomly selected images did not end up being acquired during AL. This corresponds to data unseen during training but originating from the same distribution source. For the known distribution, both methods produce low uncertainty for the majority of the test samples, as expected. However, for the single MC-Dropout network the distribution is characterized by fatter tails (both extremely confident and extremely uncertain about a significant number of images). The ensemble method, however, results in a more clustered distribution of the uncertainty. This further illustrates that ensemble learns a more representative part of the input space.

On the unseen distribution (Figure 4), the broad uniform distribution of uncertainty from the single network illustrates the presence of images about which the classifier is both extremely certain and

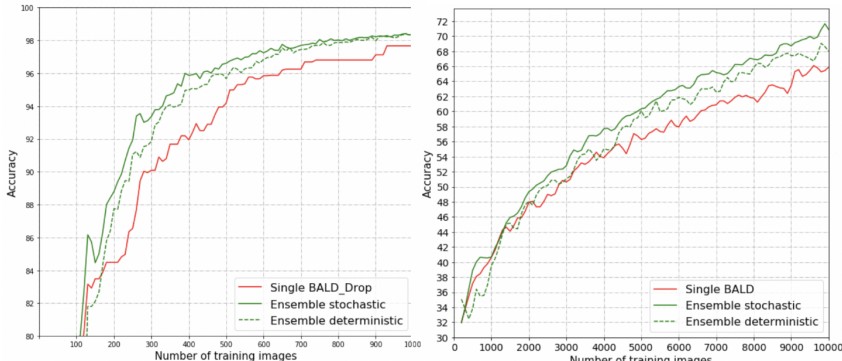

Figure 3: Test accuracy as a function of size of the incremental training set during AL. Effect of using an ensemble of three similar models (stochastic or deterministic) instead of one single MC-Dropout network. **Left:** MNIST. **Right:** CIFAR-10

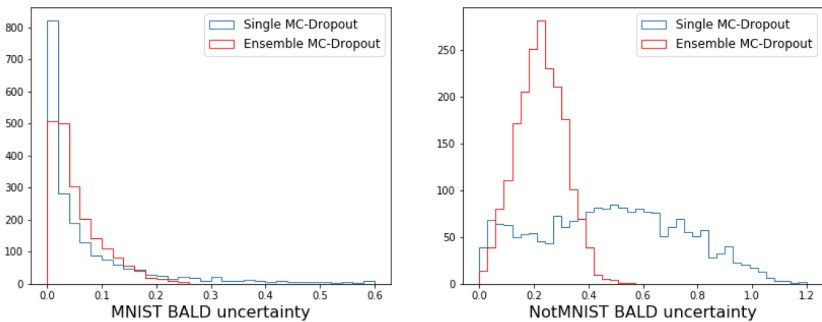

Figure 4: Histogram of BALD uncertainty of **MNIST** (left) and **NotMNIST** (right) images (2,000 random but balanced test set). Uncertainty obtained from single MC-Dropout and ensemble MC-Dropout methods at the end of the AL process.

uncertain. This implies that the network learned some specific transferable features that are recognizable in part of the new dataset. For the ensemble, on the other hand, the uncertainty is much smaller and more centered on a few values. This implies that the features learned during the initial training on MNIST are more general. This behaviour is a more realistic one to expect when evaluating a similar but new dataset. Apart from correcting for the mode-collapse phenomena, the MC-Dropout ensemble also does a better job in identifying and acquiring images from the pool set that are inherently more difficult to assign to a particular class (Appendix Figure 11).

### 3.4 DETERMINISTIC VS STOCHASTIC ENSEMBLE

In order to explain the additional improvement in DEBAL, we performed an analysis on both seen (MNIST) and unseen (NotMNIST) distributions similar to the one presented before. Figure 5 compares the histograms of BALD uncertainty obtained from the two methods using the ensemble models obtained at the end of the AL process. Additionally, we show the accuracy of the models corresponding to each binned subset of the test data. When the images are coming from a known distribution (MNIST), for both methods the accuracy decreases as the level of uncertainty increases. This observation suggests that the ambiguity captured by these methods is meaningful. However, the stochastic ensemble is more confident. Judging by the accuracy along the bins, this additional confidence seems to reflect meaningful uncertainty.

By observing the uncertainty behaviour on the unseen distribution (Figure 5, right), the stochastic ensemble is more confident overall than its deterministic counterpart, but at the same time, its uncertainty is more meaningful, as evidenced by the reduction in classification accuracy as we move towards the uncertain (right) tail of the distribution. On the other hand, the classification accuracy

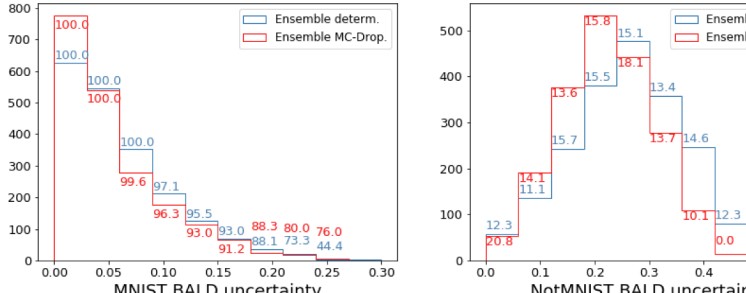

Figure 5: Histogram of BALD uncertainty of **MNIST** (left) and **NotMNIST** (right) images (2,000 random but balanced test set). Uncertainty obtained from deterministic and MC-Dropout ensemble methods at the end of the AL process. Numbers correspond to accuracy for corresponding binned subset of test data (in percentage).

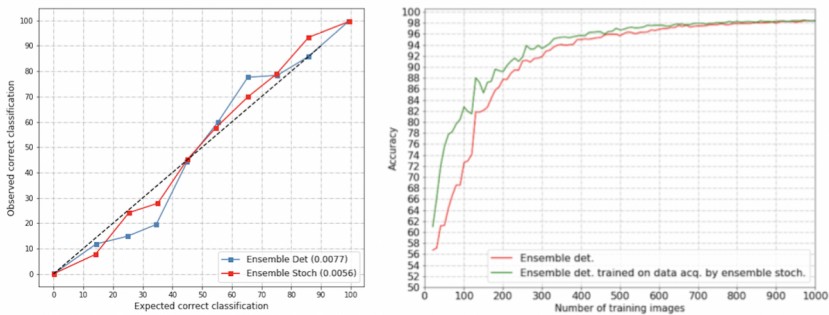

Figure 6: **Left** MNIST uncertainty calibration. Expected fraction and observed fraction. Ideal output is the dashed black line. MSE reported in paranthesis. Calibration averaged over 3 different runs.

of the deterministic ensemble is more uniform, with both tails of the distributions (most and least certain) seeing similar levels of accuracy. This suggests that the uncertainty produced by the deterministic ensemble is less correlated with the level of its uncertainty and hence less meaningful.

**Uncertainty calibration**. We used the ensemble models obtained at the end of the AL experiments to evaluate the entire MNIST test set. We looked whether the expected fraction of correct classifications matches the observed proportion. The expected proportion of correct classifications is derived from the models confidence. When plotting expected against observed fraction, a well-calibrated model should lie very close to the diagonal. Figure 6(left) shows that the stochastic ensemble method leads to a better calibrated uncertainty. An additional measure for uncertainty calibration (quality) is the Brier score (Brier, 1950), where a smaller value corresponds to better calibrated predictions. We find that the stochastic ensemble has a better quality of uncertainty (Brier score: 0.0244) compared to the deterministic one (Brier score: 0.0297). Finally, we investigated the effect of training the deterministic ensemble with data acquired by the stochastic one. Figure 6 (right) shows that incorporating stochasticity in the ensemble via MC-Dropout leads to an overall increase in performance, further reinforcing our hypothesis.

## 4 CONCLUSION AND FUTURE WORK

In this work, we focused on the use of active learning in a deep learning framework for the image classification task. We showed empirically how the mode collapse phenomenon is having a negative impact on the current state-of-the-art Bayesian active learning method. We improved upon this method by leveraging off the expressive power and statistical properties of model ensembles. We linked the performance improvement to a better representation of data uncertainty resulting from our method. For future work, this superior uncertainty representation could be used to address one of the major issues of deep networks in safety-critical applications: adversarial examples.

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

# Appendices

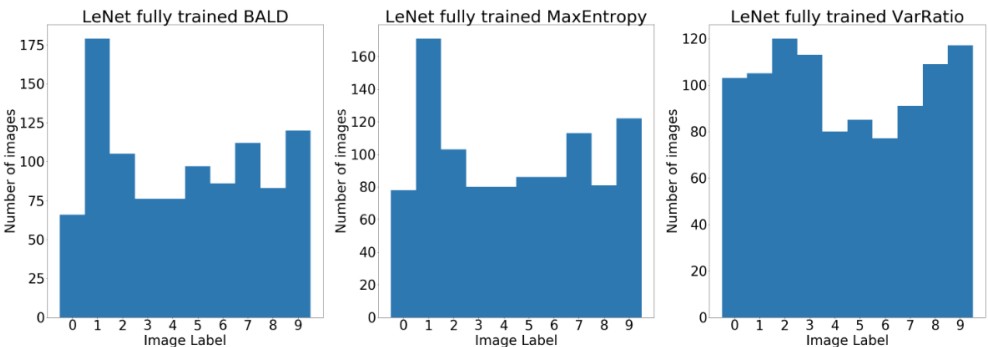

Figure 7: **MNIST** histograms of the top 1,000 most uncertain samples from test set as ranked by the LeNet model trained on the entire training set.

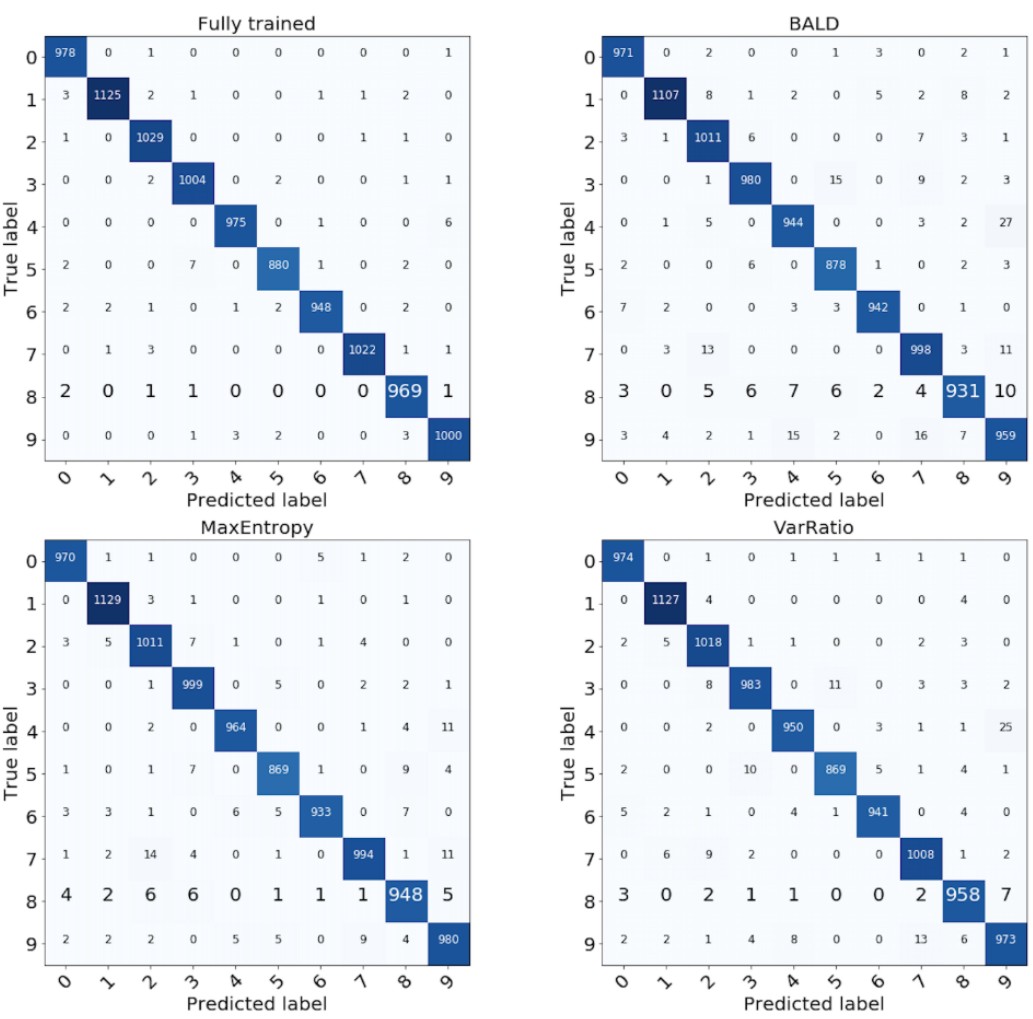

Figure 8: **MNIST** confusion matrix for the models at the end of the AL process. Test set: 10,000. Additionally, the fully trained model (top left) is shown as baseline.

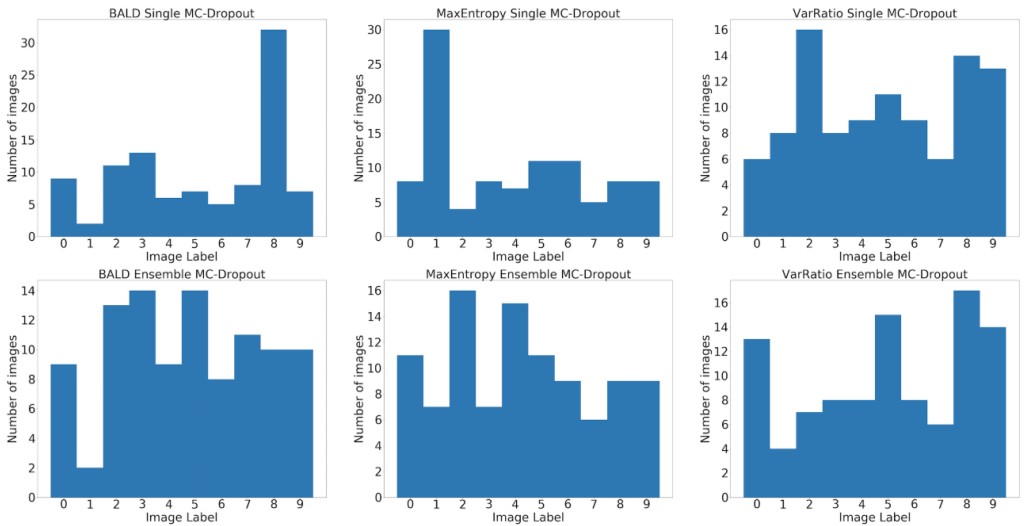

Figure 9: **MNIST** histogram of true labels in the training set after 8 acquisition iterations. Total number of images in training set: 100 **Top**: Single MC-Dropout network. **Bottom**: Ensemble of three networks of similar architecture but different random initialization.

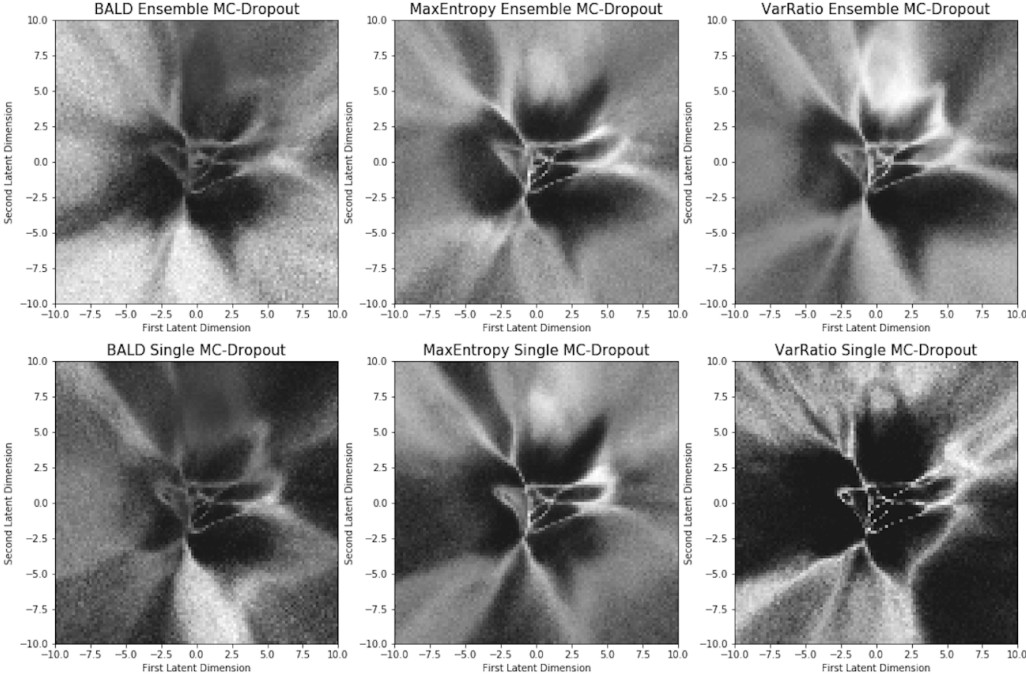

Figure 10: Uncertainty visualization in latent space. **MNIST** dataset removed for a clearer visualization of the uncertainty. Uncertainty is in white (a lighter background corresponds to higher uncertainty while a darker one represents regions of lower uncertainty) **Top:** Uncertainty obtained at the end of the AL process using an ensemble of three similar networks. **Bottom**: Uncertainty obtained at the end of the AL process using a single network.

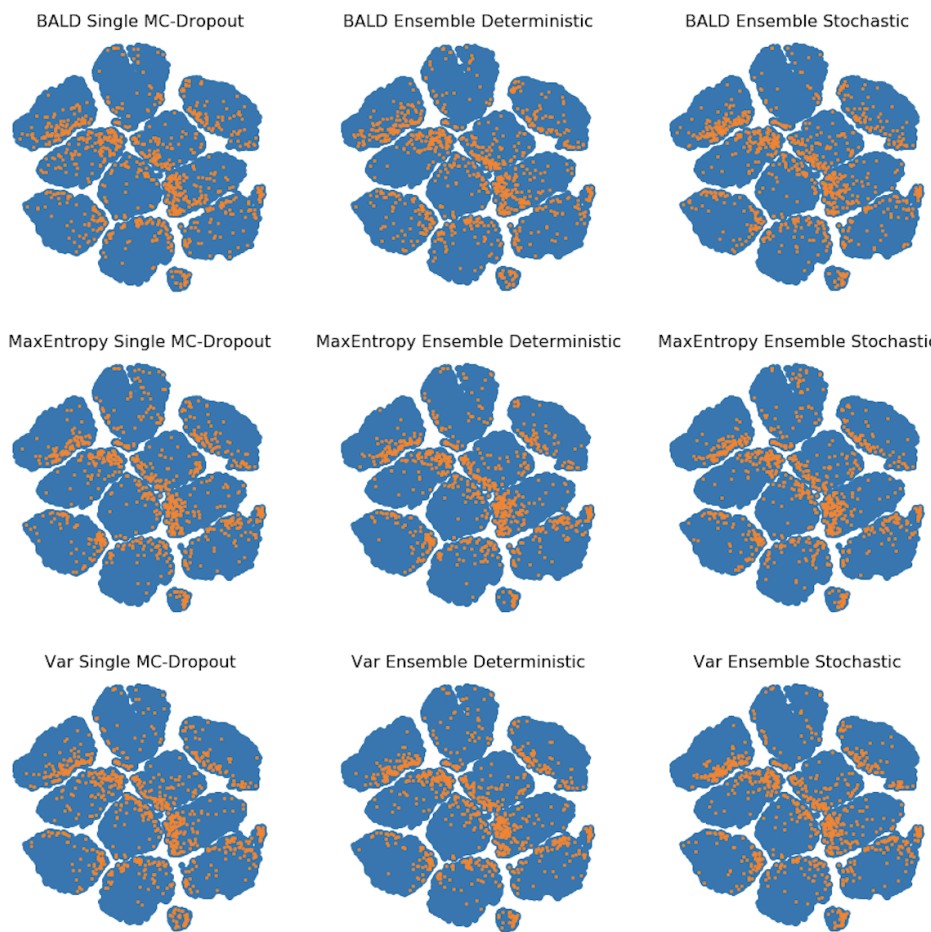

Figure 11: tSNE embeddings of the **MNIST** dataset. Effect of using an ensemble of three similar models (stochastic or deterministic) instead of one single MC-Dropout network. Orange points correspond to images acquired during the AL process.

