# OpenReview forum: "Deep Ensemble Bayesian Active Learning : Adressing the Mode Collapse issue in Monte Carlo dropout via Ensembles"
_ICLR.cc/2019/Conference_

### Official Review · AnonReviewer3 · 2018-10-31
**Clear writing but only mild improvement for computational cost.**

**Rating:** 5
**Confidence:** 4

**Review:**

This paper introduces a technique using ensembles of models with MC-dropout to perform uncertainty sampling for active learning.

In active learning, there is generally a trade-off between data efficiency and computational cost. This paper proposes a combination of existing techniques, not just ensembling neural networks and not just doing MC dropout, but doing both. The improvements over basic ensembling are rather minimal, at the cost of extra computation. More specifically, the data efficiency (factor improvement in data to achieve some accuracy) of the proposed method over using a deterministic ensemble is around just 10% or so. On the other hand, the proposed algorithm requires 100x more forward passes when computing the uncertainty (which may be significant, unclear without runtime experiments). As a concrete experiment to determine the importance, what would be the accuracy and computational comparison of ensembling 4+ models without MC-dropout vs. 3 ensembled models with MC-dropout? At the point (number of extra ensembles) where the computational time is equivalent, is the learning curve still better?

The novelty of this method is minimal. The technique basically fills out the fourth entry in a Punnett square.

The paper is well-written, has good experiments, and has a comprehensive related work section.

Overall, this paper is good, but is not novel or important enough for acceptance.

---

> ### Author Response · Authors · 2018-11-18
> **Accuracy and uncertainty for small dataset took priority over computational time**
>
> We thank our third reviewer for his comment.
>
> We do understand your concern about the significant increase in computational time. However, we believe that in the context of active learning, the main problem is not related to computational power, rather to the scarcity of data. Therefore, a better way of making the most out of little data is critical. For example, a 10 \% increase for only 300 samples acquired, could make a huge difference in a critical field where active learning is most valuable. We believe that this is exactly what we manage to achieve with our method and this comes as a result of a better representation of uncertainty during AL.
>
> Furthermore,  Beluch et al. (2018) showed that going beyond 3 networks in their deterministic ensemble method does not add any significant improvements in terms of performance. Therefore we use 3 stochastic ensembles for our method.
>
> As for the novelty of this method, although it seems more like an engineering solution, we believe that it makes a significant contribution in the field of deep active learning.

---

> > ### Comment · AnonReviewer3 · 2018-12-03
> > **Response**
> >
> > I agree that in the context of active learning, computational time is not the main constraint. However, I think if you're going to use more computational power for your method, then you should make the baselines have more computational power. In general, one would expect from a theoretical standpoint that ensembling more models will improve performance. Beluch et al. (2018) state "The performance of the ensemble-based approach is only slightly impacted by the number of members", which implies that performance was impacted, just slightly. One could argue that the impact of the MC-dropout on top of ensembling is only slight as well. I'm still not convinced of the importance or novelty of this method.

---

### Official Review · AnonReviewer1 · 2018-11-02
**Paper contains only little novelty and the experiments are not sufficiently thorough**

**Rating:** 4
**Confidence:** 4

**Review:**

The paper shows that Bayesian neural networks, trained with Dropout MC (Gal et al.) struggle to fully capture the posterior distribution of the weights.
This leads to over-confident predictions which is problematic particularly in an active learning scenario.
To prevent this behavior, the paper proposes to combine multiple Bayesian neural networks, independently trained with Dropout MC, to an ensemble.
The proposed method achieves better uncertainty estimates than a single Bayesian neural networks model and improves upon the baseline in an active learning setting for image classification.


The paper addresses active deep learning which is certainly an interesting research direction since in practice, labeled data is notoriously scarce.

However, the paper contains only little novelty and does not provide sufficiently new scientific insights.
It is well known from the literature that combining multiply neural networks to an ensemble leads to better performance and uncertainty estimates.
For instance, Lakshminarayanan et al.[1] showed that Dropout MC can produce overconfident wrong prediction and, by simply averaging prediction over multiple models, one achieves better performance and confidence scores. Also, Huand et al. [2] showed that by taking different snapshots of the same network at different timesteps performance improves.
It would also be great if the paper could related to other existing work that uses Bayesian neural networks in an active learning setting such as Bayesian optimization [3, 4] or Bandits[5].


Another weakness of the paper is that the empirical evaluation is not sufficiently rigorous:

1) Besides an comparison to the work by Lakshminarayanan et. al, I would also like to have seen a comparison to other existing Bayesian neural network approaches such as stochastic gradient Markov-Chain Monte-Carlo methods.

 2) To provide a better understanding of the paper, it would also be interesting to see how sensitive it is with respect to the ensemble size M.

 3) Furthermore, for the experiments only one neural network architecture was considered and it remains an open question, how the presented results translate to other architectures. The same holds for the type of data, since the paper only shows results for image classification benchmarks.

 4) Figure 3: Are the results averaged over multiple independent runs? If so, how many runs did you perform and could you also report confidence intervals? Since all methods are close to each other, it is hard to estimate how significant the difference is.




[1] Simple and Scalable Predictive Uncertainty Estimation using Deep Ensembles
Balaji Lakshminarayanan, Alexander Pritzel, Charles Blundel
NIPS 2017

[2] Gao Huang and Yixuan Li and Geoff Pleiss and Zhuang Liu and John E. Hopcroft and Kilian Q. Weinberger
    Snapshot Ensembles: Train 1, get {M} for free}
    ICLR 2017

[3] Bayesian Optimization with Robust Bayesian Neural Networks
    J. Springenberg and A. Klein and S.Falkner and F. Hutter
    NIPS 2016

[4] J. Snoek and O. Rippel and K. Swersky and R. Kiros and N. Satish and N. Sundaram and M. Patwary and Prabhat and R. Adams
    Scalable Bayesian Optimization Using Deep Neural Networks
    ICML 2015

[5] Deep Bayesian Bandits Showdown: An Empirical Comparison of Bayesian Deep Networks for Thompson Sampling
    Carlos Riquelme, George Tucker, Jasper Snoek
    ICLR 2018

---

> ### Author Response · Authors · 2018-11-18
> **Paper novelty, experiments and alternative methods**
>
> We thank our second reviewer for his comments. We first refer to your main comments and then answer each point in part.
>
> The work of Lakshminarayanan et al. indeed showed that deterministic ensembles can improve on the performance of MC-dropout techniques and provides a foundation for ours. And as Beluch et al. (2018) showed, this can be valuable in an active learning setting. However, our work differs in two major ways:
>
> i) We focus on showing the uncertainty representation in these methods suffer from overconfident predictions and that combining the two methods into a stochastic ensemble can be of great benefit and improve on the quality of the uncertainty.
>
> ii) We believe the true novelty to be in applying them in an active learning setting, and in particular on a small dataset problem (i.e. the size of the final dataset acquired during AL is only a small fraction of the entire available unlabelled dataset). As you mentioned, data is notoriously scarce and deep learning methods rarely work on small dataset problems.
>
> We thank the reviewer for pointing us to the work of Huand et al. Indeed this is an interesting method that would allow us to most likely achieve similar or better results with less computational overhead. This is definitely something we will consider for future work, but it is somehow out of the main scope of the paper, which was to show the power of combining MC-dropout with ensembles in the active learning setting. Taking into account more advanced ensemble methods is definitely of interest.
> In terms of the Bayesian Optimization literature, this is definitely of interest if we are to focus on hyper-parameter tuning for our models, but we fail to see the connection of the work you mentioned to our active learning examples. Our focus was not on fine-tuning our models.
>
> In relation to your specific points, we answer these below:
>
> 1) Gal has already showed in his PhD thesis that MC-Dropout almost always performs best in terms of prediction accuracy and uncertainty quality assessment when compared to alternative Bayesian neural network approaches such as Probabilistic Back Prop and other variants of stochastic gradient MCMC methods. The aim of our paper was to improve upon MC-Dropout in the context of active learning, which would invariably translate into better performance w.r.t. other Bayesian NN approaches.
> 2) Beluch et al. (2018) showed that going beyond 3 networks in their deterministic ensemble method does not add any significant improvements in terms of performance. Therefore, we used this number when benchmarking against their method.
> 3) The aim of the paper was to improve upon the state-of-the-art in active learning for the image classification task. We specifically chose this task due to its relevance to the real world especially in the medical imaging industry. We agree that a more comprehensive study could be done in order to asses the viability of our method for ML tasks other than image classification. As for other neural network architectures, we chose the one used in the benchmarked methods.
> 4) Results are averaged over 5 multiple independent runs. We will include both this and confidence scores in a revised version of our paper.

---

> > ### Comment · AnonReviewer1 · 2018-11-19
> > **response to novelty, related work and experiments**
> >
> >
> > * About stochastic vs deterministic ensembles *
> >
> > I do not understand why the work by Lakshminarayanan et al. is considered to be a deterministic ensemble technique. It uses random initialization of the neural network parameters as well as random shuffling of the data points to obtain diverse models (see Section 2.4 in the corresponding paper). Furthermore, Lakshminarayanan et al. already mentioned that one also can use other common ensemble techniques such as bagging, however it might lead to suboptimal behavior.
> >
> >
> > * About the novelty *
> >
> > The paper only shows that the mode collapse happens for Dropout MC, which apparently leads to overconfident predictions and that ensembles help to cure this. As already said above, this has been investigate by others before (Lakshminarayanan et al.).
> >
> > The novel part of the paper, compared to Beluch et al. which showed that ensembles of neural network perform better than a Bayesian neural network trained with Dropout MC, is to also train the individual ensemble components with Dropout MC. I still think the novelty is not sufficient for acceptance and that the paper would be much more convincing if the authors could present a comparison to these existing methods.
> >
> >
> > * About related work *
> >
> > Sorry, I was not very clear about that. What I mean is that there are other active learning settings such as Bayesian optimization or Bandits, where one learns a model while collecting data.  Previous work has also explored to use Bayesian neural networks in these settings, but, on hindsight, this is arguably only loosely connected to this approach.

---

> > > ### Author Response · Authors · 2018-11-19
> > > **clarifications**
> > >
> > > 1.
> > > The reason the ensembling technique he describes is deterministic is because the method he describes "We treat the ensemble as a uniformly-weighted mixture model and combine the predictions.." is
> > > using the average of M models each trained with different initializations.
> > > The random shuffling he talks about I believe has to do with how the classifier is fed the data, which in the case of  active learning this is done progressively via the the acquisition function, i.e. BALD or Entropy.
> > >
> > > Initialising a NN with random weights (say w ~ N(0,1)) will always yield the same result, given we set the same seed for our random number generator.
> > > Then averaging over 3 such models (our ensemble number M), will always yield the same result given we know what our seeds are.
> > >
> > > Now compare this to MC-dropout, where you are sampling a binary vector from a Bernoulli distribution (your dropout mask) and applying this to each layer of your NN. This is stochastic because you’re **sampling** a binary vector from a Bernoulli distribution with parameter p_i = 0.3 for example. (dropout probability of 30%). And by sampling we mean, we construct a Bernoulli distribution with a known seed, from which we sample binary variables for the hidden units, corresponding to the probability of that unit being 'on' or 'off'.
> > >
> > > What Lakshminarayanan et al. did was to compare their ensembling method to MC-dropout for regression and classification, but not for an active learning scenario and not by combining both methods and finally not for a small dataset problem (and as stated before, by that we mean, starting with very little labelled examples)
> > >
> > >
> > > 2.
> > > I would say that Beluch et al. showed that uncertainty can be better quantified via ensembles *in the active learning case* whereas what Lakshminarayanan et al. showed was that ensembles can give better uncertainty than dropout.
> > >
> > > We proposed to combine the two for the active learning scenario and small dataset problem and prove superiority in both accuracy and uncertainty in comparison to both independent evaluation, ensembles OR MC-dropout.  We were inspired in our analysis by the approach of Lakshminarayanan et al. of using the Brier score for assessing the uncertainty quality (and found out that our stochastic ensemble does have a better Brier score than the deterministic ensemble) as well as looking at the classification accuracy on unseen dataset/distributions (NotMNIST)
> > >
> > > So we did compare to both approaches (Lakshminarayanan et al. and Beluch et al.),  I'm not sure which method you would like us to compare against?

---

> > > > ### Comment · AnonReviewer1 · 2018-11-21
> > > > **response to clarifications**
> > > >
> > > > 1. Sorry I do not follow the argument. Obviously, fixing the seed of the random number generator leads to a deterministic output of the neural network, but this also holds for dropout. As one can fix the seed for sampling mini-batches or initializing the weights, one can also fix the seed for sampling from the Bernoulli distribution which leads to the exact same dropout mask. Besides that, why does one want to fix the seed for the method by Lakshminarayanan et al. at the first place? So the question why the method by Lakshminarayanan et al. is to be considered a deterministic ensemble technique remains open.
> > > >
> > > >
> > > > 2. The paper only compares an ensemble of neural networks where each networks is initialized with different random weights to an ensemble of neural networks where each networks is trained with Dropout MC. It is true that the first method resembles the work by Lakshminarayanan et al. (even though it is not clear whether they use the same proper scoring rule). However, one of Lakshminarayanan et al. main contributions was also to show that one gets better ensembles if one uses adversarial training rather than just random initializations. That should also be possible for the setting here right?
> > > >
> > > > Further more there also other baseline ensemble techniques, such as bootstrapping (e g. https://pdfs.semanticscholar.org/dde4/b95be20a160253a6cc9ecd75492a13d60c10.pdf) or the  already above mentioned snapshot ensembles. Again, due to the little novelty, I believe at least a thorough comparison to existing ensemble methods would make the paper stronger.

---

> > > > > ### Author Response · Authors · 2018-11-26
> > > > > **thank you for your comments**
> > > > >
> > > > > 1. Again, the main difference between the classical ensembles, where one initializes the networks with random initializers (as done by Lakshminarayanan et al. and us as well) and the dropout approach is that our approach of dropout is *not* deterministic. That mask is sampled, so it will have a mask with some given probability.
> > > > > "With dropout, we sample binary variables for every input
> > > > > point and for every network unit in each layer (apart from
> > > > > the last one). Each binary variable takes value 1 with probability
> > > > > pi for layer i. A unit is dropped (i.e. its value is set
> > > > > to zero) for a given input if its corresponding binary variable
> > > > > takes value 0. We use the same values in the backward
> > > > > pass propagating the derivatives to the parameters."
> > > > >
> > > > > Furthermore, I think one of the best explanations is given in this relatively new paper: https://arxiv.org/pdf/1805.09208.pdf
> > > > >
> > > > > 2. Yes adversarial training is very much on our future work, so is comparing with other ensemble methods. We thank again for your reviews, we hope to include these in a future version of the paper.

---

### Official Review · AnonReviewer2 · 2018-11-02
**Ensemble of MC-Dropout models is not an approximation of the posterior**

**Rating:** 4
**Confidence:** 4

**Review:**

The authors propose to use the combination of model ensemble and MC dropout in Bayesian deep active learning. They empirically show that there exists the mode collapse problem due to the MC dropout which can be regarded as a variational approximation. The authors introduce an ensemble of MC-Dropout models with different initialization to remedy this mode collapse problem.

The paper is clearly written and easy to follow. It is interesting to empirically show that the mode collapse problem of MC-Dropout is important in active learning.

The major concern I have is that the ensemble of MC-Dropout models is not an approximation of the posterior anymore. Each MC-Dropout model is an approximation of the posterior, but the ensemble of them may not. Therefore, it is a little misleading to still call it Bayesian active learning. Also, the ensemble of MC-Dropout models does not have the theoretic support from the Bayesian perspective.

The motivation for the proposed method is to solve the mode collapse problem of MC-Dropout, but using ensemble loses the Bayesian support benefit of MC-Dropout. So it seems not a reasonable solution for the mode collapse problem of MC-Dropout. It is not clear to me why we need to add MC-Dropout to the ensemble. What is the benefit of DEBAL over an ensemble method if both of them do not have Bayesian theoretic support?

In terms of the empirical results, the better performance of DEBAL compared to a single MC-Dropout model is not supervising as Beluch et al. (2018) already demonstrated that an ensemble is better than a single MC-Dropout. While the improvement of DEBAL compared to an ensemble is marginal but is reasonable.

The labels of figures are hard to read.

---

> ### Author Response · Authors · 2018-11-18
> **Ensemble and Bayesian dropout as posterior approximation**
>
> We thank the reviewer for its valuable and insightful comments.
>
> We are reviewing our work from a theoretical point of view and will update the paper very soon to reflect this.
>
> Even though we have not yet proved the above, we have empirically showed that the benefit of DEBAL over plain ensemble methods consists of a better representation of uncertainty, that is paramount in active learning. By better we mean
> 1) more meaningful and closer to what one would expect (Fig 4 & Fig 6 (right))
> 2) better calibrated (Fig 6 (left)). Our initial aim was not to compare stochastic ensembles with deterministic or single MC-dropout but to correct for the mode collapse issue in estimating posteriors with MC-dropout. We have empirically shown that adding ensembles to this, greatly improves the MC-dropout technique and outperforms the deterministic ensembles as well.
> We had similar doubts about the benefit of adding MC-Dropout to an ensemble. Therefore, we contrasted the performance of DEBAL against the plain ensemble method and showed empirically that DEBAL gives rise to better measures of uncertainty.
> Finally, as we strive to make our assumptions hold theoretically, we agree that adding theoretical Bayesian support to our method is of great importance if we are to further improve the understanding of Bayesian deep learning.
>
> For your final point, although Beluch et al. (2018) showed better performance for ensembles, we have shown this in the context of a small dataset problem (i.e. the size of the final dataset acquired during AL is only a small fraction of the entire available unlabelled dataset), which we believe is more relevant to the real world cases if AL is to become a widely used method.
>
> As for the figures, we are aware of this and will try to make them more clear in a revised version.

---

> > ### Comment · AnonReviewer2 · 2018-12-07
> > **Response**
> >
> > I agree that stochastic ensemble is better than the plain ensemble empirically, but the gain is incremental and lacks theoretic support.
> >
> > Again, I think the proposed method did not align with the initial goal. Your initial aim is to correct mode collapse problem in estimating posterior but the proposed method even did not estimate the posterior from Bayesian view.
> >
> > I will be glad to see the theoretic analysis in the updated version which I believe will make the paper more convincing.

---

### Meta-Review · Area_Chair1 · 2018-12-13
**A well written paper with some interesting insights but lacking novelty**

**Confidence:** 5
**Recommendation:** Reject

**Metareview:**

The reviewers in general found the paper approachable, well written and clear.  They noted that the empirical observation of mode collapse in active learning was an interesting insight.  However, all the reviewers had concerns with novelty, particularly in light of Lakshminarayanan et al. who also train ensembles to get a measure of uncertainty.  An interesting addition to the paper might be some theoretical insight about what the model corresponds to when one ensembles multiple models from MC Dropout.  One reviewer noted that it's not clear that the ensemble is capturing the desired posterior.

As a note, I don't believe there is agreement in the community that MC dropout is state-of-the-art in terms of capturing uncertainty for deep neural networks, as argued in the author response (and the abstract).  To the contrary, I believe a variety of papers have improved over the results from that work (e.g. see experiments in Multiplicative Normalizing Flows from over a year ago).